# PROMPT OPTIMIZATION VIA ADVERSARIAL IN-CONTEXT LEARNING

## ABSTRACT

We propose a new method, Adversarial In-Context Learning (adv-ICL), to optimize prompt for in-context learning (ICL) by employing one LLM as a *generator*, another as a *discriminator*, and a third as a *prompt modifier*. As in traditional adversarial learning, adv-ICL is implemented as a two player game between the generator and discriminator, where the generator tries to generate realistic enough output to fool the discriminator. In each round, given an input prefixed by task instructions and several exemplars, the generator produces an output. The discriminator is then tasked with classifying the generator input-output pair as model-generated or real data. Based on the discriminator loss, the prompt modifier proposes possible edits to the generator and discriminator prompts, and the edits that most improve the adversarial loss are selected. We show that adv-ICL results in significant improvements over state-of-the-art prompt optimization techniques for both open and closed-source models on 11 generation and classification tasks including summarization, arithmetic reasoning, machine translation, data-to-text generation, and the MMLU and big-bench hard benchmarks. In addition, because our method uses pre-trained models and updates only prompts rather than model parameters, it is computationally efficient, easy to extend to any LLM and task, and effective in low resource settings.

## 1 INTRODUCTION

Generative Adversarial Networks (GANs) and adversarial learning (Goodfellow et al., 2014) have driven significant progress across a range of domains, including image generation (Goodfellow et al., 2014; Radford et al., 2015; Arjovsky et al., 2017), domain adaptation (Ganin et al., 2016; Tzeng et al., 2017; Xie et al., 2017; Louppe et al., 2017), and enhancing model robustness (Szegedy et al., 2013; Biggio et al., 2013; Carlini & Wagner, 2017; Madry et al., 2018). At its core, adversarial learning frames training as a minimax game between a *generator* and a *discriminator*. The generator aims to generate output realistic enough that the discriminator classifies it as real (i.e., not generated), while the discriminator aims to differentiate between generator output and training data samples as accurately as possible. After each round, the parameters of both models are updated based on an adversarial loss, and the process is repeated. As the generator improves, the discriminator improves alongside it, finding "weak spots" in generator output that may go undiscovered in a non-adversarial setup, resulting in better outputs from the generator.

Though adversarial learning has been effective in other domains, the traditional adversarial learning setup requires updating model parameters, which is highly impractical for pretraining large language models (LLMs) due to data and compute constraints. Particularly for novel tasks where data is often scarce, it is desirable to have methods that can improve model performance using limited data. In this work, we solve this problem by applying adversarial learning to *in-context learning (ICL)* (Radford et al., 2019; Brown et al., 2020; Chowdhery et al., 2022; Touvron et al., 2023a; Beltagy et al., 2022; Liu et al., 2023), keeping model parameters fixed and instead updating the prompts given to each model in an adversarial manner. This alleviates requirements on compute and data, while still improving model performance. We refer to our method as *Adversarial In-Context Learning* (adv-ICL).

adv-ICL uses an adversarial objective and three main modules (as shown in Figure 1) to optimize the prompt for a given task. Each module consists of an LLM powered by a specific prompt. The first

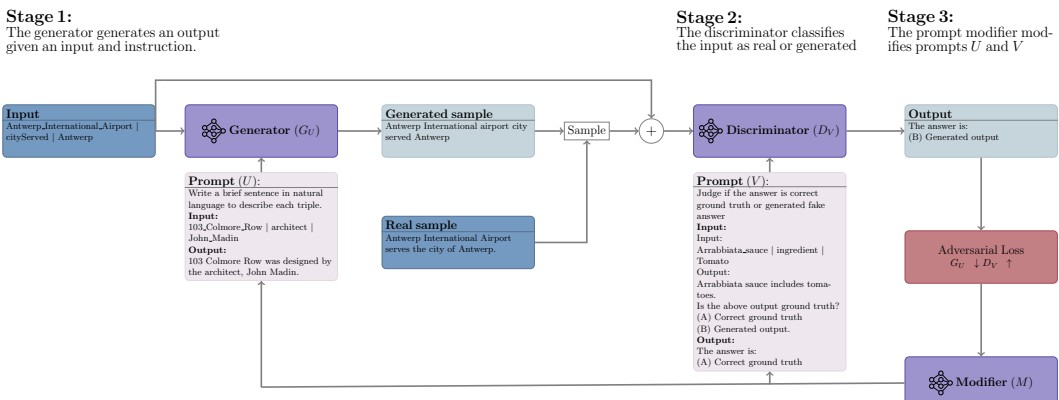

Figure 1: adv-ICL sets up a minimax game between two players: a *Generator* and a *Discriminator*, both of which are LLMs powered by few-shot prompts. The generator's inputs are examples without labeled outputs while the discriminator's inputs are examples coming from the generator and chosen ground truth. The generator's role is to produce the response, whereas the discriminator's goal is to differentiate between the ground truth and the generated answers. Both are updated using a standard adversarial objective by adjusting their respective prompts. This process is aided by another LLM, the *Prompt Modifier*, which updates prompts through a repeated sampling and selection procedure based on the adversarial loss.

module is a generator ($G$), which is tasked with generating realistic, task appropriate output given a task instruction and an input. The second is a discriminator ($D$) which has the goal of classifying inputs as real or produced by $G$. Finally, there is a prompt modifier $M$ which is responsible for updating the prompts to $G$ and $D$. As in typical adversarial learning, the learning objective is set up as a minimax game between $G$ and $D$. In each round, $G$ produces an output based on an input and a prompt consisting of a task instruction and several example inputs and outputs. $D$ then classifies the pair constructed of the original input and $G$'s output as generated or real. Finally, $M$ produces a number of possible updates to $G$ and $D$'s prompts, the updates that most improve the adversarial loss from $D$'s classification are selected, and the procedure repeats.

We evaluate adv-ICL on 13 tasks using various open and closed-source LLMs, finding that adv-ICL outperforms other state-of-the-art prompt optimization techniques by large margins across different model configurations and tasks. For instance, we improve the accuracy of ChatGPT (OpenAI, 2022) from 71.0% to 74.0% on MMLU (Hendrycks et al., 2021), 79.9% to 82.3% on GSM8K (Cobbe et al., 2021), and 72.1% to 74.0% on BBH (Suzgun et al., 2022). Importantly, adv-ICL requires very few iterations and training samples to achieve this, boosting performance significantly after only five training rounds using twenty training points. Finally, adv-ICL is also easy to implement, encouraging its use in real-world applications.

## 2 ADVERSARIAL IN-CONTEXT LEARNING

### 2.1 BACKGROUND: IN-CONTEXT LEARNING

With the scaling of model sizes (Brown et al., 2020; Chowdhery et al., 2022; Touvron et al., 2023a; OpenAI, 2023), Large Language Models (LLMs) have demonstrated strong capabilities in solving downstream tasks through conditioning only on an input prompt containing a few demonstrations (a.k.a., few-shot prompting). This paradigm is known as prompt-based learning or *in-context learning (ICL)* (Radford et al., 2019; Beltagy et al., 2022; Liu et al., 2023). ICL simplifies the process of adapting a general-purpose LLM to cater to a specific task without having to do feature engineering or model training.

Formally, given a specific task, let the LLM generator be represented by $G_U$. $G_U$ is driven by a prompt $U = (I^G, x_1^G, y_1^G, \cdots, x_k^G, y_k^G)$, where $I^G$ is the task instruction, $x_i^G$ is a sample input, and $y_i^G$ is the corresponding sample output. The generator's output for a new input $x$, then, is determined by the instruction and the exemplars in $U$, making the choice of $U$ crucial in determining the downstream performance of $G_U$ (Deng et al., 2022; Pryzant et al., 2023).

## 2.2 Adversarial Training Objective

adv-ICL optimizes the generator's prompt using an adversarial approach, inspired by GANs (Goodfellow et al., 2014)—in particular cGAN (Mirza & Osindero, 2014) and BiGAN (Donahue et al., 2016) where the discriminator deals with the conditional distribution and joint distribution of an input and output. As in GANs, it is essential to optimize both the discriminator and generator in the adv-ICL framework concurrently, to make sure that they reach a desired optimal state. Concretely, to assess the output of our generator, $G_U$, we employ a discriminator, $D_V$, which attempts to classify $G_U$'s output as real or generated.

Similar to $G_U$, $D_V$ is an LLM driven by a prompt $V = (I^D, x_1^D, y_1^D, z_1^D, \cdots, x_k^D, y_k^D, z_k^D)$, where $I^D$ is a task instruction, $x_i^D$ is a sample input, $y_i^D$ the corresponding output, and $z_i^D$ a label of real or generated representing whether $y_i^D$ is a generated example or a real data sample. $D_V$ utilizes a loss function $\mathcal{J}$ inspired by GANs, formally defined as follows:

$$\mathcal{J}(D_V, G_U) = \mathbb{E}_{x,y \sim p_{data}} \log \Big( D_V(x, y) \Big) + \mathbb{E}_{x \sim p_{data}} \log \Big( 1 - D_V \big( x, G_U(x) \big) \Big) \tag{1}$$

where $p_{data}$ is the distribution of real data. Note that, in this case, the discriminator is designed for the binary decision problem of determining whether the input is generated or real. In our prompt, we represent the choices as two options: (A) real or (B) generated. As a result, we can evaluate the classification probability based on the generation probability of option (A), where $D_V(x, y) = 1$ indicates a real sample. Therefore, in order for $G_U$ to improve its performance, its goal is for $D_V$ to mis-classify its outputs as real as often as possible (i.e. minimizing $\mathcal{J}$). In contrast, $D_V$'s objective is to increase $\mathcal{J}$, indicating improved classification ability. Formally, this adversarial training objective can be expressed as the following minimax game:

$$\min_U \max_V \mathcal{J}(D_V, G_U) \tag{2}$$

Since the discriminator is powered by a large language model with enough capacity, the optimal solution for this minimax objective indicates that the generator's output, when paired with its input, becomes indistinguishable from the real.

---

**Algorithm 1** Adversarial In-Context Learning Optimization

---

**Input:** $U = (I^G, x_1^G, y_1^G, \cdots, x_k^G, y_k^G)$, $V = (I^D, x_1^D, y_1^D, z_1^D, \cdots, x_k^D, y_k^D, z_k^D)$.
**Input:** Generator $G_U$, Discriminator $D_V$, Prompt Modifier $M$.
**Input:** #training iterations $T$, #samples used per iteration $m$, #new sampled prompts $r$.
 1: **for** $T$ training iterations **do**
 2:     Sample $m$ data points from the set of limited samples to compute $J(G_U, D_V, m)$.
 3:     `// Optimize the instruction` $I^D$ `for` $D_V$
 4:     Generate $r$ new instructions $\{I_1, I_2, ..., I_r\}$ from $I^D$ using the prompt modifier $M$.
 5:     Substitute $I_n$ to $V$ $\forall n \in \{1, 2, ..., r\}$ to compute the loss $J_n(G_U, D_V, m)$, and select the largest $J_j$.
 6:     Update $I^D$ by $I_j$ if $J_j > J$.
 7:     `// Optimize the demonstrations` $(x_i^D, y_i^D, z_i^D)$ `∀i for` $D_V$
 8:     **for** $i \in range(k)$ **do**
 9:         Generate $r$ new $((x_{i1}, y_{i1}, z_{i1}), ..., (x_{ir}, y_{ir}, z_{ir}))$ from $(x_i^D, y_i^D, z_i^D)$ using $M$.
10:         Substitute $(x_{in}, y_{in}, z_{in})$ to $V$ $\forall n \in \{1, 2, ..., r\}$ to compute the loss $J_{in}(G_U, D_V, m)$, and select the largest $J_{jn}$.
11:         Update $(x_i^D, y_i^D, z_i^D)$ by $(x_{ij}, y_{ij}, z_{ij})$ if $J_{jn} > J$.
12:     **end for**
13:     `// Similarly optimize` $U$ `for` $G_U$ `so that` $J(G_U, D_V, m)$ `decreases.`
14:     ...
15: **end for**
**Output:** The optimized prompt $U$ for the Generator $G_U$.

---

## 2.3 ADVERSARIAL IN-CONTEXT LEARNING OPTIMIZATION

Whereas GANs optimize model parameters using backpropagation, adv-ICL does not update the parameters of $G_U$ and $D_V$ directly, but instead updates their prompts in each iteration of the game. This requires a number of differences in our optimization process. First, we consider a setting where we have access only to model outputs and generation probabilities, making it impossible to use backpropagation as our method for updating $U$ and $V$. Therefore, we employ a third LLM to serve as the *prompt modifier*, $M$. Given a prompt's task instruction $I$ or demonstration $(x, y)$ as input, $M$ generates $r$ possible variations on it. The adversarial loss is recomputed for each variation by substituting the variation into the original prompt, and the output that improves the adversarial loss the most is returned as the modification, following Gonen et al. (2022).

We refer to our optimization algorithm as *Adversarial In-Context Learning Optimization*, which can been seen in pseudocode form in Algorithm 1.The entire process is as follows: Given the initial generator prompt $U$, and discriminator prompt $V$, we run $T$ training iterations. At each iteration, we first sample $m$ pairs of data points from our training samples to compute the adversarial training loss $\mathcal{J}(G_U, D_V, m)$. We then optimize the loss by using $M$ to modify both the task instruction and demonstration portions of the prompts for the discriminator and generator.

## 2.4 THEORETICAL ANALYSIS

In this section, we theoretically analyze whether such a minimax objective in the form of in-context learning can achieve the desired equilibrium as in the original GAN scenario. We assume access to models with infinite capacities powering the discriminator $D$, generator $G$, and prompt modifier $M$ and that in each iteration, we sample a sufficient number of prompts from $M$ to update both $G$ and $D$. Let $p_{data}$ be the distribution of the training data, and $p_g$ be of the generated data from $G$.

Considering a language model $\mathcal{M}$ which can be $D$ or $G$ performing a corresponding task $T_1$ and evaluated by a metric $E_1$, we further assume that:

1. $M$ is powerful enough to modify the initial prompt of $\mathcal{M}$ for $T_1$, covering all possible prompt variants performing the task $T_1$.
2. $\mathcal{M}$ is a powerful enough language model that there exists a prompt $\mathcal{P}$ of $T$ for $\mathcal{M}$ that given $\mathcal{P}$, $\mathcal{M}$ can achieve the globally optimal result on $E_1$ for the task $T$.
3. There exists a prompt sampled by $M$ that maximizes $\mathcal{M}$ on $E_1$ globally on $T$.

With the above assumptions, we prove the following results.

**Proposition 1.** *(Motivated by Goodfellow et al. (2014)) If $G$ and $D$ have enough capacity, and at each training step, the discriminator is allowed to reach its optimum $D^*$ given $G$, and $p_g$ is updated so as to improve the criterion*

$$\mathcal{J}(D^*, G) = \mathbb{E}_{x, y \sim p_{data}} \log\left(D^*(x, y)\right) + \mathbb{E}_{x \sim p_{data}} \log\left(1 - D^*\big(x, G(x)\big)\right) \tag{3}$$

*then $p_g$ converges to $p_{data}$.*

The full proof of the proposition 1 can be found in Appendix A.1. Our conclusion is that with strong enough $D, G, M$, the framework adv-ICL converges. In practice, convergence in adversarial training is a complex and challenging problem. For example, there is no universally applicable stop criterion for training GANs. Previous studies often rely on the number of iterations as a stop criterion (Goodfellow et al., 2014; Radford et al., 2015), as the standard adversarial loss function alone is inadequate for determining when to stop GAN training (Salimans et al., 2016). However, despite these challenges, our approach demonstrates significant improvements even with just a few training iterations and samples, showcasing its effectiveness, which we have extensively examined.

## 2.5 IMPLEMENTATION DETAILS

**Zero-shot Prompt Modification**   We leverage the capability of LLMs to follow human instructions to generate $r$ variations of a given task instruction/demonstration. Specifically, we use three prompt

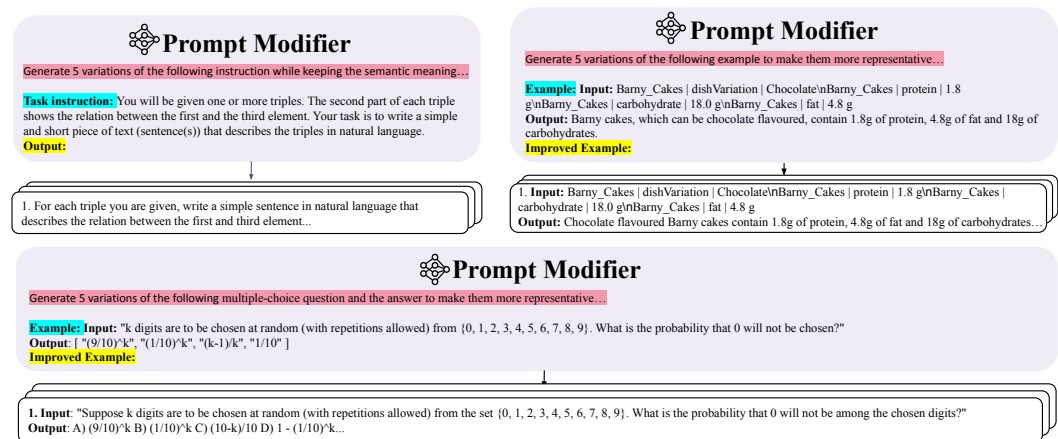

Figure 2: Examples of how the prompt modifier generates new prompts $U$ for $G_U$ including new task instructions and new data examples. The full prompts used are presented in Appendix A.3.

templates, one for generating instructions, one for generating open-ended question-answer pairs, and one for multiple-choice question pairs. We present each of them with one example with incomplete output in Figure 2.

**Hyperparameter Selection** As shown in Algorithm 1, our proposed algorithm involves three main hyperparameters: the number of training iterations $T$, the number of data points used per iteration $m$, and the number of new versions sampled for each instruction/demonstration $r$. We fix $r$ to be five. It is worth noting the larger $r$ is, the more expense is required, and the more likely improvements are gained. For $T$ and $m$, our analysis in Section 3.3 proves that given a discriminator and a generator, selecting suitable combinations of $T$ and $m$ is critical for obtaining strong performance. Therefore, we use a simple hyperparameter search method to select a good combination. First, we collect a small set $S$ of samples from the validation set of 3 representative tasks from all tasks. Next, we run the grid search algorithm for $T \in \{1, 3, 5\}$ and $m \in \{1, 2, 5, 10\}$, 12 experiments in total. Finally, we compute the performance of adv-ICL on $S$ and select the best combination as the values of $T$ and $m$ for our algorithm. We outline how we construct $S$ in Section 3.1.

## 3 EXPERIMENTATION

### 3.1 EXPERIMENTAL SETUP

**Datasets** We conduct experiments on a total of 13 NLP tasks in four main categories: *generation*, *classification*, *reasoning*, and challenging NLP *evaluation suites* to verify the effectiveness of adv-ICL. For generation, we select XSUM (Narayan et al., 2018) and CNN/Daily Mail (CNN for short) (Nallapati et al., 2016) as our *text summarization* benchmarks; WebNLG (Gardent et al., 2017) and E2E NLG (Novikova et al., 2017) as our *data-to-text generation* datasets; and LIRO (RO → EN) (Dumitrescu et al., 2021) and TED Talks (IT→ JA) (Ye et al., 2018) as our *machine translation* benchmarks. In the classification category, we use YELP-5 (Zhang et al., 2015), COPA (Roemmele et al., 2011) and WSC (Levesque et al., 2012). For reasoning tasks, GSM8K (Cobbe et al., 2021) and SVAMP (Patel et al., 2021) are chosen as arithmetic reasoning benchmarks. Finally, we also evaluate our method on two challenging *evaluation suites*: MMLU (Hendrycks et al., 2021) and BIG-bench Hard (BBH) (Suzgun et al., 2022). Due to computational and budget limitations, except for GSM8K and SVAMP, each benchmark is evaluated on a maximum of 1,000 test samples randomly chosen from the test set. In our preliminary experiments, we found that the empirical results on the sampled test set is aligned with performance on the whole test set. The exact number of testing samples for each task is presented in Appendix A.3.

One of the main advantages of in-context learning is that it is able to generalize to new tasks with limited training examples, as may be the case for novel tasks. To make our method applicable in such settings, we use 20 labeled samples for training adv-ICL. For our baseline methods, we assume access to at most 100 labeled data samples for each benchmark except BBH, similar to previous

prompt optimization works (Xu et al., 2022; Pryzant et al., 2023). For BBH, we assume access to three chain-of-thought data samples per task.

**Backbone Models**   We test state-of-the-art open and closed-source LLMs as our backbone models. For the open-sourced models, we use *Vicuna-13B v1.5* (Zheng et al., 2023) – an open-source chat model fine-tuned on top of *LLaMa 2* (Touvron et al., 2023b) via instruction fine-tuning. For our closed-source models, we use *text-davinci-002* and *ChatGPT (gpt-3.5-turbo-0613)* (OpenAI, 2022), which are built on top of *GPT-3* (Brown et al., 2020). For each backbone model except ChatGPT, we use the same model for the generator, discriminator, and prompt modifier in the adv-ICL setup. Since ChatGPT does not provide the probabilities of its generated tokens, which is required for computing the adversarial loss, we employ *text-davinci-002* as the discriminator when ChatGPT is the generator and the prompt modifier.

**Baselines**   We compare adv-ICL with five baselines: (i) Simple prompting (*Few-shot*) that is typically used. We use Chain-of-Thought (CoT) (Wei et al., 2022) for reasoning tasks; (ii) Utilizing ROUGE-L score (Lin, 2004) (*ROUGE-L*) as the criteria to optimize the instruction and demonstrations for each task on a small sampled labeled set; (iii) Similarly, using Perplexity (*Perplexity*) as the criteria following Gonen et al. (2022); (iv) Genetic Prompt Search (*GPS*) (Xu et al., 2022), a genetic optimization method based on the log-logits or accuracy; (v) Automatic Prompt Optimization (*APO*) (Pryzant et al., 2023), which uses data to generate text "gradients" evaluating the current prompt, and then utilize them to signal the models to edit the prompt in the opposite semantic direction. (vi) Automatic Prompt Engineer (APE) (Zhou et al., 2022), which automatically generates instructions and selects via evaluation scores.[1]

We make sure that all methods use a similar number of labeled samples, while the exact number of training samples depends on the design of specific algorithms. For GPS and APO, we sample 32 and 50 labeled data examples for validation, following (Xu et al., 2022; Pryzant et al., 2023). For ROUGE-L and Perplexity, we sample 80 data examples for validation. For YELP, WSC, GSM8K, SVAMP, where the benchmarks do not have enough labeled examples, we sample from their limited training set instead. Additionally, APO requires additional training data for error samples. For fair comparisons, we use the same training data with adv-ICL. More implementation details for baselines are presented in Appendix A.2.

**Prompt Initialization**   We follow prior works to employ a set of initialized prompts. For MMLU and BBH, we employ the open-sourced prompts that come with the original papers. For GSM8K and SVAMP, we follow the chain-of-thought paper Wei et al. (2022) which employs human-written prompts. For the remaining benchmarks, we utilize prompts from Super-NaturalInstructions (Wang et al., 2022), in which instructions and demonstrations are chosen by domain experts. All the initial prompts are also used for our baseline *few-shot* experiments. The exact number of shots used for each benchmark is presented in Appendix A.3.

**Evaluation Metrics**   For the generation tasks, we evaluate the performance of the frameworks by ROUGE-L score (Lin, 2004), following Wang et al. (2022). For classification tasks, we use accuracy as the evaluation metric. For MMLU and BBH, we follow Hendrycks et al. (2021); Suzgun et al. (2022) and report the averaged performance among tasks.

**Hyperparameters**   To select a set of appropriate hyperparameters for adv-ICL, we use one representative task in each category. Specifically, we use WebNPL for generation, GSM8K for reasoning, and MMLU for classification. The selected hyperparameters are then used for all the tasks. We test the performance of our method on a sampled validation set for these three tasks, refer to this set as S (Section 2.3). We use 80 data samples from WebNPL and 80 data samples of GSM8K[2]. For MMLU, $16, 16, 17, 19$ samples from the validation sets of `abstract_algebra`, `business_ethics`, `econometrics`, `formal_logic` are selected respectively, resulting in 228 samples in S.

After hyperparameter selection, we set number of training iterations $T$ to 3 and number of training samples per iteration $m$ to 5 for all the tasks except BBH. For BBH, we set $T = 3$, $m = 3$ given that

---

[1]As APE only polishes task instruction, we compare APE with Adv-ICL on GSM8K, MMLU and WebNLG.
[2]GSM8K does not come with a validation set, so we sample from the training set instead.

we only have 3 samples in the training set. In each iteration, the prompt modifier samples $r = 5$ new prompts. We discuss the details of hyperparameter search in Section 3.3.

## 3.2 MAIN RESULTS

We present the main empirical results on a set of classification, generation and reasoning tasks in Table 1, MMLU in Table 2, and BBH in Figure 3.

| Models | Method | Summarization | | Data-to-Text | | Translation | | Classification | | | Reasoning | |
|---|---|---|---|---|---|---|---|---|---|---|---|---|
| | | XSUM | CNN | WebNLG | E2E NLG | LIRO | TED Talks | YELP Review | COPA | WSC | GSM8K | SVAMP |
| text-davinci-002 | Few-shot | 25.5 | 20.8 | 60.8 | 47.1 | 78.3 | 37.7 | 71.1 | 87.9 | 67.7 | 47.3 | 70.0 |
| | ROUGE-L | 25.8 | 21.1 | 61.1 | 47.5 | 77.6 | 38.2 | 70.6 | 87.8 | 66.9 | 47.1 | 69.8 |
| | Perplexity | 26.2 | 21.4 | 62.2 | 49.3 | 78.5 | 39.0 | 70.9 | 88.6 | 67.3 | 47.5 | 70.4 |
| | GPS | 27.1 | 21.5 | 61.9 | 49.1 | 78.8 | 39.4 | 71.3 | 87.4 | 67.1 | 48.1 | 70.5 |
| | APO | 26.8 | 22.1 | 62.3 | 49.2 | 78.9 | 40.2 | 71.1 | 88.8 | 68.3 | 46.9 | 69.3 |
| | adv-ICL | 30.9↑3.8 | 23.4↑1.3 | 65.4↑3.1 | 50.8↑1.5 | 81.2↑2.3 | 42.1↑1.9 | 74.4↑3.1 | 92.2↑3.4 | 73.8↑5.5 | 50.8↑2.7 | 72.5↑2.0 |
| Vicuna v1.5 | Few-shot | 18.9 | 16.4 | 52.5 | 35.3 | 72.1 | 32.6 | 71.0 | 77.8 | 54.4 | 40.7 | 45.1 |
| | ROUGE-L | 18.9 | 16.6 | 52.7 | 35.2 | 72.6 | 32.9 | 70.9 | 76.7 | 54.1 | 40.4 | 44.8 |
| | Perplexity | 19.1 | 16.9 | 52.8 | 35.0 | 72.7 | 33.0 | 71.0 | 77.9 | 54.7 | 41.4 | 46.2 |
| | GPS | 19.7 | 16.9 | 53.0 | 35.9 | 73.2 | 33.0 | 71.3 | 78.2 | 55.0 | 41.7 | 45.7 |
| | APO | 19.5 | 17.1 | 53.7 | 36.3 | 73.1 | 32.9 | 70.2 | 78.3 | 54.4 | 41.4 | 46.3 |
| | adv-ICL | 21.1↑1.4 | 19.3↑2.2 | 59.3↑5.6 | 41.9↑5.6 | 73.4↑0.2 | 35.2↑2.2 | 73.6↑2.3 | 81.6↑3.3 | 58.2↑3.2 | 43.9↑3.2 | 48.4↑3.3 |
| ChatGPT | Few-shot | 25.2 | 21.3 | 60.9 | 48.3 | 78.8 | 41.7 | 69.8 | 94.4 | 69.8 | 79.4 | 79.3 |
| | ROUGE-L | 25.1 | 21.2 | 60.7 | 48.6 | 78.5 | 41.3 | 68.2 | 93.7 | 69.1 | 78.7 | 78.9 |
| | Perplexity | 24.9 | 20.9 | 61.8 | 48.6 | 78.9 | 41.8 | 68.8 | 91.3 | 66.9 | 75.5 | 78.1 |
| | GPS | 26.6 | 21.5 | 61.5 | 48.9 | 78.9 | 42.0 | 70.0 | 94.6 | 69.8 | 79.4 | 80.0 |
| | APO | 27.1 | 22.1 | 61.5 | 49.3 | 79.4 | 42.3 | 70.3 | 94.8 | 70.1 | 79.9 | 79.7 |
| | adv-ICL | 28.2↑1.1 | 22.5↑0.4 | 63.6↑1.8 | 51.1↑1.8 | 80.4↑1.0 | 43.2↑0.9 | 71.9↑0.6 | 95.8↑1.0 | 71.9↑1.8 | 82.3↑2.4 | 81.1↑1.1 |

Table 1: Main experimental results on generation, classification and reasoning tasks. Details of the selected few-shot prompts and the baselines are described in Section 3.1.

**Generation Tasks** As shown in Table 1, adv-ICL significantly outperforms all the baseline methods across all backbone models, achieving $2.3\%, 2.9\%, 1.2\%$ absolute improvements on average for text-davinci-002, Vicuna and ChatGPT respectively. We observe that adv-ICL achieves most significant improvements on Summarization and Data-to-Text tasks. Specifically, for *text-davinci-002*, adv-ICL outperforms the best baseline by 3.8% on XSUM and 3.1% on the WebNLG data-to-text task. For Vicuna v1.5, adv-ICL achieves an improvement of 5.6% on the two data-to-text generation tasks WebNLG and E2E NLG. For ChatGPT, we achieve an improvement of 3.0% on XSUM and 2.8% on the E2E NLG generation task when compared to the vanilla few-shot baseline where no prompt optimization is applied. When compared to other prompt optimization methods, we hypothesize that the smaller but respectable improvements on ChatGPT may be due to the misalignment between the backbone models of the generator and the discriminator. However, given that ChatGPT is the most widely used LLM, undergoing constant upgrades to better serve millions of people daily, it should be expected that improving ChatGPT is more difficult.

**Classification Tasks** For classification tasks, adv-ICL also brings significant improvements over all the SOTA prompt optimization techniques across all the models with $4.0\%, 2.9\%, 0.8\%$ absolute improvements on average respectively. Specifically, the most significant performance improvement is obtained with the `text-davinci-002` backbone. The 2.9% improvements on Vicuna also illustrates the effectiveness of our proposed method on open-sourced models. The improvements of the three backbone models on the three classification tasks are relatively balanced.

**Reasoning Tasks** For reasoning tasks, we observe a $2.7\%$ and $2.0\%$ absolute improvement on GSM8K and SVAMP, with text-davinci-002. Likewise, significant gains are observed with ChatGPT, achieving a $2.4\%$ increase on GSM8K and a $1.1\%$ boost on SVAMP. In the case of Vicuna, it achieves $3.2\%$ absolute improvement on GSM8K and $3.3\%$ absolute improvement on SVAMP. The effectiveness of adv-ICL for reasoning tasks, particularly when coupled with CoT prompting, where the prompt includes detailed intermediate reasoning steps, demonstrates its ability to optimize complex prompts. This hints at potential for applying adv-ICL to more advanced prompting methods.

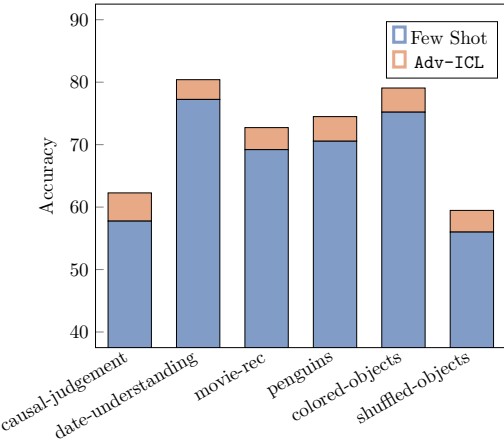 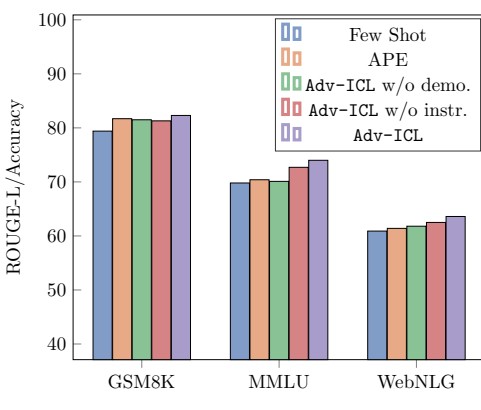

Figure 3: Results on selected tasks from BBH with ChatGPT using 5-shot Chain-of-Thought prompting. We achieve an average accuracy of 70.6% while the baseline method achieves an average of 68.2%. Full results can be found in Appendix A.5

Figure 4: Ablation study on ChatGPT with adv-ICL in which we only update the task instruction or demonstrations.

| Models | Method | Humanity | STEM | Scocial Sciences | Others | Avg |
|--------|--------|----------|------|------------------|--------|-----|
| Vicuna v1.5 | Few-shot | 55.8 | 38.7 | 63.3 | 61.5 | 54.6 |
| | ROUGE-L | 55.5 | 39.5 | 63.7 | 61.1 | 55.0 |
| | Perplexity | 55.2 | 39.5 | 64.1 | 61.9 | 55.2 |
| | GPS | 56.9 | 40.4 | 64.1 | 62.3 | 55.9 |
| | APO | 57.2 | 40.0 | 63.7 | 62.7 | 55.9 |
| | adv-ICL | **58.9** ↑1.7 | **44.1** ↑3.7 | **64.8** ↑0.7 | **64.5** ↑1.8 | **58.1** ↑2.2 |
| ChatGPT | Few-shot | 73.9 | 57.5 | 79.2 | 73.5 | 71.0 |
| | ROUGE-L | 74.2 | 56.7 | 78.4 | 73.9 | 70.8 |
| | Perplexity | 74.8 | 56.3 | 79.6 | 71.2 | 70.5 |
| | GPS | 74.6 | 57.9 | 80.0 | 74.3 | 71.7 |
| | APO | 75.6 | 58.3 | 80.7 | 73.9 | 72.1 |
| | adv-ICL | **76.7** ↑1.1 | **61.3** ↑3.0 | **82.3** ↑1.6 | **75.8** ↑1.5 | **74.0** ↑1.9 |

Table 2: Results of ChatGPT using 5-shot prompts on MMLU.

**MMLU & BBH**   We summarize the results on MMLU in Table 2. We improve the average performance from 69.8% to 73.1%, achieving performance improvements on 51 subjects out of 57 subjects with ChatGPT. For BBH, as shown in Figure 3, adv-ICL achieves an accuracy of 70.6% where the baseline method achieves an accuracy of 68.2% with ChatGPT and chain-of-thought prompting. The detailed results on MMLU and BBH are in Appendix A.5.

Note that for BBH, only three data examples are provided with the dataset. Consequently, we use the same three examples as the initial data for both the generator and discriminator. Additionally, these 3 examples are the only real data examples utilized when estimating the objective. Despite this, we achieve substantial improvements on this task. This demonstrates the broad applicability of our method. In real-world scenarios, where the number of training examples is relatively limited, our approach can still be effectively applied.

## 3.3  FURTHER STUDIES

In this section, we examine several design choices of adv-ICL. We further discuss the necessity of the discriminator in Appendix A.4, as well as an extended set of analyses in Appendix A.5.

**Optimizing task instruction / demonstration only**   As instruction and demonstration data are both widely used in prompts, we examine the importance of optimizing these two components separately. We use ChatGPT in these experiments and compare our method with another prompt optimization

| $m \setminus T$ | $T = 1$ | $T = 3$ | $T = 5$ |
|---|---|---|---|
| $m = 1$ | 61.3 / 78.8 / 42.6 | 63.8 / 80.0 / 47.1 | 62.5 / 80.0 / 48.5 |
| $m = 3$ | 62.5 / 81.3 / 45.6 | 65.0 / 81.3 / 52.9 | 62.5 / 76.3 / 50.0 |
| $m = 5$ | 63.8 / 82.5 / 54.4 | **66.3 / 82.5 / 55.9** | 63.8 / 77.5 / 54.4 |
| $m = 10$ | 60.0 / 80.0 / 51.5 | 62.5 / 81.3 / 51.5 | 63.8 / 78.8 / 47.1 |

(a) ChatGPT (G) & text-davinci-002 (D).

| $m \setminus T$ | $T = 1$ | $T = 3$ | $T = 5$ |
|---|---|---|---|
| $m = 1$ | 52.5 / 40.0 / 50.0 | 53.8 / 43.8 / 55.9 | 53.8 / 42.5 / 54.4 |
| $m = 3$ | 55.0 / 42.5 / 48.5 | 60.0 / 43.8 / 54.4 | 57.5 / 45.0 / 51.5 |
| $m = 5$ | 55.0 / 41.4 / 48.5 | **61.3 / 45.0 / 54.4** | 57.5 / 42.5 / 51.5 |
| $m = 10$ | 53.8 / 42.5 / 52.9 | 55.0 / 42.5 / 50.0 | 55.0 / 41.3 / 45.6 |

(b) Vicuna (G) & Vicuna (D).

Table 3: Ablation studies on number of iterations $T$ and number of samples used per iteration $m$. The results are ROUGE-L / Acc / Acc scores on WebNLG / GSM8K / MMLU.

method, APE (Zhou et al., 2023), on three tasks: WebNLG, GSM8K (with CoT) and MMLU. The results are shown in Figure 4.

First, we see that updating the instruction only, or the demonstrations only makes the model perform suboptimally. Second, optimizing demonstrations are more effective than optimizing instructions for WebNLG and MMLU while the situation is the opposite on GSM8k. We hypothesize that this is because generated reasoning chains can contain errors and the correctness of the generated answers with respect to the generated questions is critical for the model's performance (Min et al., 2022). That said, adv-ICL still achieves significant performance improvements in both cases for GSM8k.

**Human evaluation of prompt modifier performance** The capability of the prompt modifier to follow human instructions to update the prompts is crucial for our proposed method. In our experiments, text-davinci-002, ChatGPT, and Vicuna are used in the zero-shot prompting manner (Figure 2). To evaluate their capabilities in modifying the prompts following the instructions, we hire three annotators per backbone model to manually rate 100 generated cases (30 instructions, 70 demonstrations) as Satisfied / Unsatisfied. Since a human annotator could not tell if an instruction or demonstration would lead to better results, we ask annotators to label a case as Satisfied when the sampled instruction / demonstration is semantically similar to the original one, and Unsatisfied otherwise. We observe that three models achieved strong Satisfied rates with 88%, 91%, and 83% for text-davinci-002, ChatGPT, and Vicuna respectively. The Unsatisfied cases are mostly observed from sampling the demonstrations. More detailed results are available in Appendix A.5.

**Ablation studies on number of iterations $T$ and data samples $m$** As discussed in Section 3.1, we perform hyperparameter search with three datasets including WebNLG, GSM8k and MMLU. We conduct the experiments with two ChatGPT and Vicuna as the backbone models. As shown in Table 3, we observe the best performance achieved with $T = 3$ and $m = 5$ for both settings. This demonstrates that our method works effectively without requiring many training iterations and data samples. We further provide our explanations regarding training with too many iterations $T$ or samples $m$ might harm the performance of the models in .

### 3.4 QUALITATIVE ANALYSIS

To intuitively understand how the optimization goes, we show how prompts change over iterations in Figure 5 for data-to-text generation task WebNLG. The prompt modifier significantly alters the generator's prompt. In two iterations, it initially simplifies the instruction and then adds a more specific requirement. The demonstrations are either replaced with a completely new one or are refined.

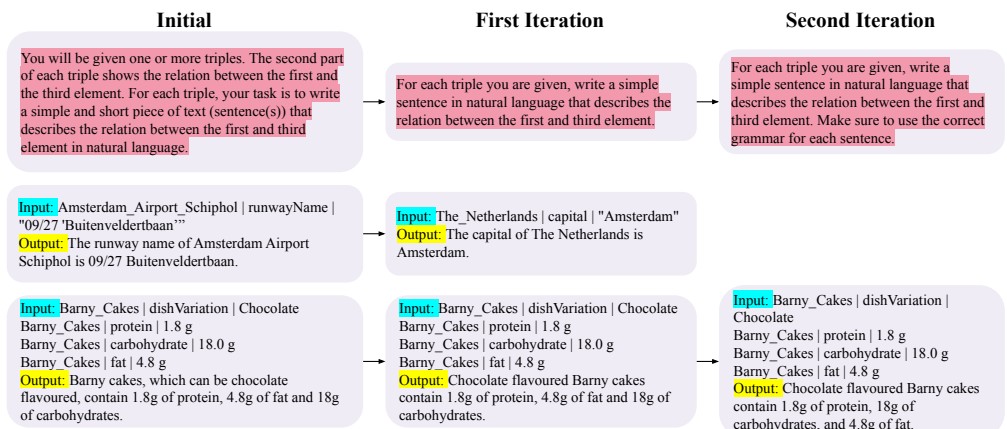

Figure 5: Optimization for the prompt on the data-to-text task WebNLG.

# 4 RELATED WORK

**Adversarial Training** Adversarial training has been widely used in image generation (Goodfellow et al., 2014; Radford et al., 2015; Arjovsky et al., 2017), domain adaptation (Ganin et al., 2016; Tzeng et al., 2017; Xie et al., 2017; Louppe et al., 2017), and improving model robustness (Szegedy et al., 2013; Biggio et al., 2013; Carlini & Wagner, 2017; Madry et al., 2018). However, previous work shows that it often harms the generalization of models (Raghunathan et al., 2019; Min et al., 2021). In NLP, there is an increasing interest in adversarial training; however, most of the current research primarily examines its effect on generalization (Cheng et al., 2019; Wang et al., 2019; Jiang et al., 2020), and finetuning the models (Jin et al., 2020; Liu et al., 2020), which is impractical for recent gigantic language models. In contrast, adv-ICL targets to optimize the prompts and demonstrates strong generalization under different conditions.

**Prompt Optimization** The emergence of in-context learning (Radford et al., 2019; Brown et al., 2020; Chowdhery et al., 2022; Touvron et al., 2023a; OpenAI, 2023) has sparked interest in prompt optimization (PO) techniques (Qin & Eisner, 2021; Deng et al., 2022; Lu et al., 2022; Xu et al., 2022; Pryzant et al., 2023; Yang et al., 2023), which can lead to substantial performance gained for LLMs. Previous PO works can be cast into two different types of prompts: (1) continuous prompts; and (2) discrete textual prompts. Some notable works optimizing continuous prompts such as (Qin & Eisner, 2021; Liu et al., 2021; Lester et al., 2021). However, as model sizes increase, this approach becomes more computationally expensive. In the context of very large language models, recently, Xu et al. (2022) propose a gradient-free prompt optimization method called Genetic Prompt Search (GPS) by iteratively generating prompts and selecting the top-K ones in each iteration. In addition, Pryzant et al. (2023) introduce Automatic Prompt Optimization (APO) leveraging text "gradients" to evaluate the current prompt, and then using them to modify the prompt in the opposite semantic direction. In this work, we compare adv-ICL with GPS and APO. We notice other prompt optimization techniques such as Automatic Prompt Engineer (Zhou et al., 2023) optimizing only the task instructions, which is also compared with a variant of adv-ICL. There are also RL-based prompt optimization baselines such as (Deng et al., 2022; Lu et al., 2022). However, we exclude RL-based methods from our comparison because they involve training additional MLPs and lack a universal reward.

# 5 CONCLUSION

In this work, we introduce adv-ICL, an adversarial training framework for in-context learning using large language models. Our method has demonstrated empirical success across a diverse range of tasks and outperforms previous SOTA prompt optimization methods significantly. Requiring only limited data samples and a very small number of training iterations, adv-ICL holds promise for implementation in a wide array of real-world applications.

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

# A  APPENDIX

## A.1  THEORETICAL PROOFS OF THE CONVERGENCE

In this section, we theoretically analyze whether such a minimax objective in the form of in-context learning can achieve the desired equilibrium as in the original GAN scenario. We assume access to models with infinite capacities powering the discriminator $D$, generator $G$, and prompt modifier $M$ and that in each iteration, we sample a sufficient number of prompts from $M$ to update both $G$ and $D$. Let $p_{data}$ be the distribution of the training data, and $p_g$ be the distribution of the generated data from $G$.

Considering a language model $\mathcal{M}$ which can be $D$ or $G$ performing a corresponding task $T_1$ and evaluated by a metric $E_1$, we further assume that:

1. $M$ is powerful enough to modify the initial prompt of $\mathcal{M}$ for $T_1$, covering all possible prompt variants performing the task $T_1$.
2. $\mathcal{M}$ is a powerful enough language model that there exists a prompt $\mathcal{P}$ of $T$ for $\mathcal{M}$ that given $\mathcal{P}$, $\mathcal{M}$ can achieve the globally optimal result on $E_1$ for the task $T$.
3. There exists a prompt sampled by $M$ that maximizes $\mathcal{M}$ on $E_1$ globally on $T$.

The assumption 3 is a result of assumptions 1, and 2, and the assumption about our access to infinite capacities language models. Indeed, given $\mathcal{M}$, from assumption 2, there exists a globally optimized prompt $\mathcal{P}$ of $T_1$ for it such that it can achieve the globally optimal state on $E_1$ for the task $T_1$. Furthermore, since $M$ is powerful enough in modifying the initial prompt (ass. 1), plus $M$ samples a sufficiently large number of prompts for each iteration (ass. 2), $M$ can generate $\mathcal{P}$ with a non-zero probability, which conclude the assumption 3.

With the above assumptions, we prove the following results.

**Proposition 2.** *(Goodfellow et al., 2014) For $G$ fixed, the optimal discriminator $D$ can be described in a closed form, denoted as $D^*$.*

*Proof for Proposition 2, adapted from (Goodfellow et al., 2014).* For a fixed $G$, the training objective for the discriminator $D$ is maximizing the adversarial loss $\mathcal{J}(D, G)$ (Equation (1))

$$
\begin{aligned}
\mathcal{J}(D, G) &= \mathbb{E}_{x,y \sim p_{data}} \log \Big( D(x, y) \Big) + \mathbb{E}_{x \sim p_{data}} \log \Big( 1 - D\big(x, G(x)\big) \Big) \\
&= \mathbb{E}_{x,y \sim p_{data}} \log \Big( D(x, y) \Big) + \mathbb{E}_{x,y \sim p_g} \log \Big( 1 - D\big(x, y\big) \Big) \\
&= \int_x p_{data}(x) \log D(x, y) \Big) dx + \int_x p_g(x) \log \Big( 1 - D\big(x, y\big) \Big) dx \\
&= \int_x p_{data}(x) \log D(x, y) + p_g(x) \log \Big( 1 - D\big(x, y\big) \Big) dx
\end{aligned}
\tag{4}
$$

The function $y = a \log(x) + b \log(1 - x)$ for $(a, b) \in \mathbb{R}^2 and (a, b) \neq \{0, 0\}$ achieves its maximum in $[0, 1]$ at $\frac{a}{a+b}$. Therefore, $D^*(x)$ has a closed form, which is $D^*(x) = \frac{p_{data}(x)}{p_{data}(x) + p_g(x)}$. $\square$

**Proposition 3.** *For each training iteration, for a fixed $G$, the optimal discriminator $D^*$ can be achieved.*

*Proof for Proposition 3.* From assumption 3, at each training iteration, by fixing $G$ and taking the adversarial loss function $\mathcal{J}(D, G)$ as $E_1$, there exists a prompt sampled by $M$ that maximizes $D$ on $E_1$ globally, which constitutes to $D^*$. $\square$

**Proposition 4.** *(Motivated by Goodfellow et al. (2014)) If $G$ and $D$ have enough capacity, and at each training step, the discriminator is allowed to reach its optimum $D^*$ given $G$, and $p_g$ is updated so as to improve the criterion*

$$\mathcal{J}(D^*, G) = \mathbb{E}_{x,y \sim p_{data}} \log \left( D^*(x,y) \right) + \mathbb{E}_{x \sim p_{data}} \log \left( 1 - D^*\big(x, G(x)\big) \right) \qquad (5)$$

*then $p_g$ converges to $p_{data}$.*

*Proof for Proposition 4.* At each training step, from proposition 3, the optimal $D^*$ can be achieved via editing its input prompt by $M$. Considering the loss function $\mathcal{J}(D^*, G)$ as a function in $p_g$, then $\mathcal{J}(D^*, G)$ is convex in $p_g$. Since $G$ is powerful enough that there exists a prompt $\mathcal{P}$ sampled by $M$ such that $G$ can achieve the globally optimal loss $\mathcal{J}$ (assumption 2), with an optimal $D^*$, we can obtain the corresponding best $G$. Furthermore, $\mathcal{J}(D^*, G)$ is convex in $p_g$, plus the global optimal of $G$ can be obtained, with a sufficiently large enough number of prompts sampled and training iterations, $p_g$ converges to $p_{data}$. □

In practice, it is impossible that 1, 2, 3 assumptions could be achieved, plus our approach is based on GAN, which is theoretically known to lack guaranteed convergence, our proposed framework naturally does not guarantee convergence either. Convergence in GANs is a complex and challenging problem, and there is no universally applicable stop criterion for training GANs. Previous studies often rely on the number of iterations as a stop criterion (Goodfellow et al., 2014; Radford et al., 2015), as the standard adversarial loss function alone is inadequate for determining when to stop GAN training (Salimans et al., 2016). However, despite these challenges, our approach demonstrates significant improvements even with just a few training iterations and samples, showcasing its effectiveness, which we have extensively examined.

### A.2  BASELINE IMPLEMENTATION

In this section, we present our implementation details for the baselines. First, among the benchmarks we used, the following datasets do not have any validation set with sizes larger than or equal to 80: YELP, WSC, GSM8K, SVAPM. Therefore, we randomly sample 100 data cases from their training sets, to create their validation sets.

Each baseline requires a development set to decide which prompt(s) is/are the best at each optimization iteration. For GPS and APO, we sample 32 and 50 data samples respectively from the validation set of each benchmark, following Xu et al. (2022); Pryzant et al. (2023). For ROUGE-L and Perplexity, we sample 80 data samples, also from each validation set. Additionally, among the baselines, only APO requires training data for error messages. For a fair comparison with adv-ICL, we use the same training data samples with adv-ICL as training data for APO.

● **ROUGE-L & Perplexity (Gonen et al., 2022)**  For these baselines, we utilize ROUGE-L (Lin, 2004) or Perplexity Gonen et al. (2022) as the measurement to optimize the input instruction and demonstrations sequentially. For the instruction, we sample 15 new instructions by paraphrasing following the template: `'Write for me 15 paraphrases of the {initial_instruction}:'`. We then select the version which achieves the best result on $S$ as the final instruction. Similarly, for each demonstration, we use the template `'Write 15 paraphrases for the following example. Keep the format as Input: and Output:. End the answer by So the answer is:'` to sample 15 versions of the original demonstrations, and select the best one on $S$ sequentially until all the demonstrations are optimized. We sample 15 versions for comparisons because our proposed adv-ICL also samples a maximum of 15 versions for the instruction and each demonstration.

● **GPS (Xu et al., 2022)**  We run GPS (Xu et al., 2022) on 3 iterations to optimize the instruction and each demonstration sequentially. Denote the original instruction/demonstration to be optimized as $O$. In the initial step, given the original human-written $O$, we paraphrase it into 10 versions using `'Write for me 10 paraphrases of the {initial_instruction}:'` for instruction, and `'Write 10 paraphrases for the following example. Keep the format as Input: and Output:. End the answer with <END>. So the answer is:'` for demonstration. We then select the top-5 generated $O$ to pass to the first iteration. At each iteration, for each $O$ in the current top$-5$ $Os$, we sample 5 new $Os$ by Sentence Continuation strategy (Schick & Schütze, 2021) via using the backbone LLM itself, and select the

top$-5$ $Os$ among $25$ $Os$ to the next iteration. Finally, the best-performing $O$ on $S$ is selected as the output instruction/demonstration of the method. It is worth noting that in the original paper from Xu et al. (2022), top$-k$ with $k = 25$ was used. However, in our reimplementation, we use $k = 5$ so that it can be relatively fair to compare GPS with our method (we use $r = 5$) and other baselines. The template for sampling new prompts via the Sentence Continuation strategy that we used is exactly the same as Xu et al. (2022) provided.

• **APO (Pryzant et al., 2023)**   Since our setting assumes that we have access to limited training data samples, we reimplemented a simplified version of the original APO in which the selection step (Pryzant et al., 2023) only be called once, and the samples that we used to train adv-ICL are returned. For simplicity, we call the original instruction/demonstration as $O$. We run APO to optimize the instruction and each demonstration sequentially in a given prompt. Given an initial $O$, and the error samples, we use the backbone LLM to generate feedback consisting of $5$ comments as the text "gradient". Integrating this gradient as feedback, we ask the LLM to generate $10$ prompt samples. We further utilize the backbone LLM to generate $5$ paraphrase versions of the original $O$, resulting in a total of $15$ new $Os$. Finally, we select the best $O$ evaluated on $S$. All the prompt templates for generating gradients, integrating feedback, and generating paraphrased prompts are adopted from Pryzant et al. (2023). For selecting error samples, in the original implementation, Pryzant et al. (2023) compared the generated answer with the ground-truth answer, and the error samples are the ones that have the generated answer different from the ground-truth answers. This is applicable for classification and numerical question-answering tasks, but not the text generation tasks such as summarization, this strategy of selecting error samples is not suitable. Therefore, for summarization, data-to-text, and translation tasks, we select one sample that the current prompt brings the lowest ROUGE-L score as the sole error sample.

• **APE (Zhou et al., 2023)**   For APE, we adopt the implementation on the GitHub[3] from Zhou et al. (2023). We limit the number of instructions sampled to $15$ to have fair comparisons with adv-ICL. For the training samples for each task, we use the same samples that we train adv-ICL for APE.

## A.3   SUPPLEMENTARY EXPERIMENT DETAILS

In this section, we provide more details used in the experiments.

**Number of demonstrations for few-shot experiments**   Number of demonstrations for few-shot experiments of all datasets is listed in Table 4. For generation tasks and classification tasks, We follow the expert-written prompts from Super-NaturalInstruction (Wang et al., 2022). For reasoning tasks, MMLU and BBH, we follow the standard prompts that they propose in their paper or open-source code.

|  | Summarization | | Data-to-Text | | Translation | | Classification | | | Reasoning | | Evaluation Suits | |
|---|---|---|---|---|---|---|---|---|---|---|---|---|---|
|  | XSUM | CNN | WebNLG | E2E NLG | RO $\rightarrow$ EN | IT$\rightarrow$ JA | YELP Review | COPA | WSC | GSM8K | SVAMP | MMLU | BBH |
| #shots | 3 | 2 | 3 | 2 | 3 | 3 | 3 | 3 | 3 | 5 | 5 | 5 | 3 |

Table 4: Number of shots used for *few-shot* experiments.

**Test set Statistics**   As mentioned in the main paper, we sample a subset of the test set for efficient evaluation. In Table 5, we show the exact numbers of testing samples we used for each task.

|  | Summarization | | Data-to-Text | | Translation | | Classification | | | Reasoning | |
|---|---|---|---|---|---|---|---|---|---|---|---|
|  | XSUM | CNN | WebNLG | E2E NLG | RO $\rightarrow$ EN | IT$\rightarrow$ JA | YELP Review | COPA | WSC | GSM8K | SVAMP |
| #test samples | 1000 | 950 | 1000 | 1000 | 1000 | 1000 | 1000 | 496 | 285 | 1319 | 1000 |

Table 5: Test set statistics.

**Prompt Modifier prompts**   Here, we also provide the prompt used in the prompt modifier. The prompt is as follows:

---
[3]https://github.com/keirp/automatic_prompt_engineer/tree/main

- Modifying instructions: `Generate 5 variations of the following instruction while keeping the semantic meaning. Keep the generated instructions as declarative. Wrap each with <START> and <END>.`.

- Modifying open-ended QA pairs: `Generate 5 variations of the following example to make them more representative. Keep the format as Input: and Output:. Wrap each with <START> and <END>.`.

- Modifying MCQ pairs: `Generate 5 variations of the following multiple-choice question and the answer to make them more representative. Keep the format as multiple-choice question and the answer. Keep the format as Input: and Output:. Wrap each with <START> and <END>.`.

**Extended experimental details**   For OpenAI API models, ChatGPT (gpt-3.5-turbo-0613) with chat completion mode and text-davinci-002 with text completion mode were called at temperature 0.6. For open-source baselines, Vicuna v1.5 13B was used with a window size of 1024. We use Nucleus Sampling Holtzman et al. (2020) as our decoding strategy for all the models with a p value of 0.9.

## A.4   WHY THE DISCRIMINATOR WORKS?

We further conduct experiments (Table 6) to verify whether the prompt modifier module work as expected. Specifically, we remove the discriminator and only employ a prompt modifier to repeatedly optimize the prompt.

|  | WebNLG | RO $\rightarrow$ EN | YELP | GSM8K |
|---|---|---|---|---|
| Vicuna 13B | 52.5 | 72.1 | 71.0 | 40.7 |
| adv-ICL *w.o. discriminator* | 50.1 | 71.4 | 72.1 | 40.2 |
| adv-ICL | 59.3 | 73.4 | 73.6 | 43.9 |
| ChatGPT | 60.9 | 78.8 | 69.8 | 79.4 |
| adv-ICL *w.o. discriminator* | 61.2 | 77.4 | 64.5 | 71.6 |
| adv-ICL | 63.6 | 80.4 | 71.9 | 82.3 |

Table 6: Experimental results with Vicuna and ChatGPT with adv-ICL when being removed the discriminator.

In most cases, removing the discriminator and relying solely on the prompt modifier under Vicuna and ChatGPT leads to a decline in performance. This observation highlights the importance of the discriminator and adversarial loss in the optimization process.

## A.5   EXTENDED EXPERIMENTS

**Reliability of the results**   We rerun our experiments with adv-ICL three times on WebNLG, RO $\rightarrow$ EN, YELP, GSM8K. The results are presented in Table 7.

|  | WebNLG | RO $\rightarrow$ EN | YELP | GSM8K |
|---|---|---|---|---|
| Vicuna 13B | 59.3/59.2/59.5 | 73.4/74.1/73.2 | 73.6/73.6/73.5 | 43.9/44.3/44.1 |
| ChatGPT | 63.6/63.5/63.8 | 80.4/80.6/80.6 | 71.9/71.8/71.9 | 82.3/82.5/82.2 |

Table 7: Our experimental results with adv-ICL on three different runs.

The results clearly demonstrate that adv-ICL consistently delivers stable outcomes, thereby highlighting its reliability in faithfully reproducing our experimental findings.

**Providing more feedback to the prompt modifier**   We conducted an experiment that involved integrating the most successful prompts from previous iterations as feedback for the next iteration. In this process, we utilized previous best-performing prompts, namely $P_1, P_2, ..., P_k$, as inputs to the prompt constructor module in order to generate the $(k + 1)$-th prompt, denoted as $\{P_1, ..., P_k\}$. The template for optimizing task instruction is shown as follows, similar to the prompt for optimizing demonstrations.

```
Diversify the task instruction to be clearer.  Keep the task
instruction as declarative.
Task instruction:  P_0
Improved task instruction:  P_1
...
Task instruction:  P_{k-1}
Improved task instruction:  P_k
Task instruction:  P_k
Improved task instruction:
```

We applied the method to four representative tasks WebNLG, RO → EN, YELP, GSM8K using both Vicuna and ChatGPT models. The obtained results for are illustrated in Table 8.

| | WebNLG | RO → EN | YELP | GSM8K |
|---|---|---|---|---|
| Vicuna 13B | 52.5 | 72.1 | 71.0 | 40.7 |
| adv-ICL (prompt modifier with history) | 56.9 | 74.0 | 74.2 | 42.2 |
| adv-ICL | 59.3 | 73.4 | 73.6 | 43.9 |
| ChatGPT | 60.9 | 78.8 | 69.8 | 79.4 |
| adv-ICL (prompt modifier with history) | 62.1 | 79.8 | 72.1 | 80.9 |
| adv-ICL | 63.6 | 80.4 | 71.9 | 82.3 |

Table 8: Experimental results with Vicuna and ChatGPT with the feedback to the prompt modifier.

In the case of Vicuna, incorporating additional feedback into the prompt modifier proves effective for tasks such as translation and classification. However, this approach falls short when applied to data-to-text and reasoning tasks. On the other hand, for ChatGPT, augmenting the prompt modifier with more feedback does not yield improved performance. This can be attributed to ChatGPT's strong zero-shot prompt capabilities, which outshine its ability to perform effectively with few-shot prompts.

**Ablation studies on number of generated samples** $r$   We investigate whether generating fewer / more samples in each prompt modification would affect the model's performance. Due to the limited resources, we only conducted the experiment on the WebNLG and GSM8k dataset, with $r \in \{1, 3, 5, 10, 20\}$. The results are shown in Figure 6. We observe that increasing $r$ lead to comparable results.

**Why might too many iterations $T$ or samples $m$ harm the performance of the models?**   We observed this phenomenon in the experiments and were also curious about it. We hypothesize that first, training with too many iterations can cause the model to be overfitting to the task, leading to worse performance on the test samples. Second, adv-ICL, a specialized form of in-context learning, plays a crucial role in enhancing the performance of LLMs by enabling them to learn from the training examples and generate improved prompts. While in-context learning holds great promise, it is essential to acknowledge that increasing the number of training examples does not necessarily guarantee better performance. As demonstrated by Min et al. (2022), a critical threshold exists for the number of training examples, and surpassing this threshold leads to a decline in performance. Thus, in our specific settings, augmenting the training examples did not yield better results.

Given its inherent complexity and non-deterministic nature, we have put forward a hyper-parameter tuning approach, presented in Table 3, aimed at determining these hyper-parameters for new configuration settings.

**Prompt Modifier temperature**   Lastly, we examine the influence of the generation temperature for the prompt modifier. Ideally, the prompt modifier should have enough diversity to generate potential improvements for the prompts of both the generator and discriminator. Intuitively, this means we should not use greedy decoding with a temperature of 0 for the prompt modifier. As demonstrated in Figure 7, a temperature of 0.6 works well, providing a sufficiently large search space while still generating high-quality prompts.

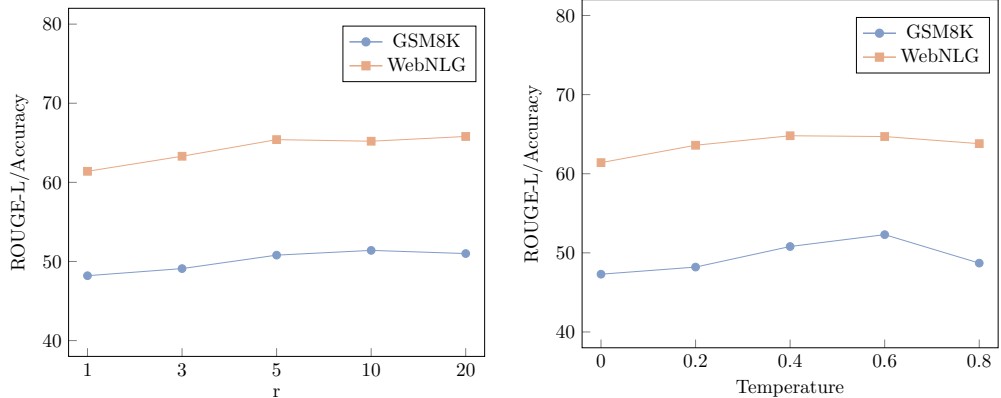

Figure 6: Ablation study on the number of sampled prompt $r$.

Figure 7: Ablation study on temperature of the prompt modifier.

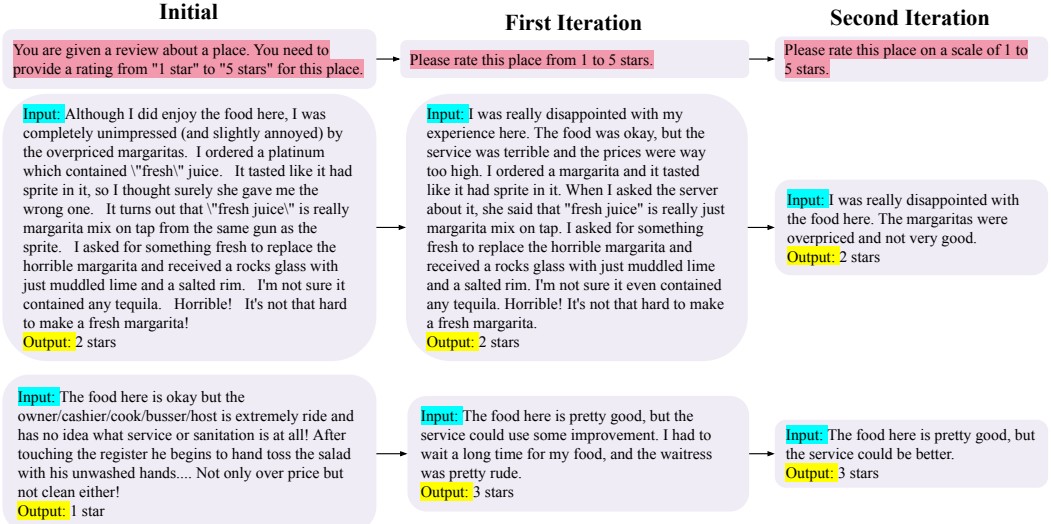

Figure 8: Qualitative analysis on the classification task Yelp.

**Prompt Modifier performance inspection**   Here we present the detailed results of human evaluation on generated instructions and demonstrations respectively. Details are shown in Table 9. text-davinci-002 and ChatGPT achieve similar performance with the zero-shot prompt modifier, while Vicuna performs a little bit worse but also achieves an acceptable correctness ($\geq 80$).

| Model | 30 instructions | 70 demonstrations | Overall |
|---|---|---|---|
| text-davinci-002 | 93.3 | 85.7 | 88.0 |
| Vicuna v1.5 | 90.0 | 80.0 | 83.0 |
| ChatGPT | **96.7** | **88.6** | **91.0** |

Table 9: Human evaluation results for each specific type of modifications.

**More qualitative analysis**   Here, we also show an additional case of qualitative analysis on Yelp. As shown in 8, the optimization follows a similar pattern with that on the data-to-text task.

**Detailed results on MMLU**   In Figure 9, we show the detailed results on MMLU with ChatGPT. As shown in the graph, adv-ICL achieves significant improvements on most tasks.

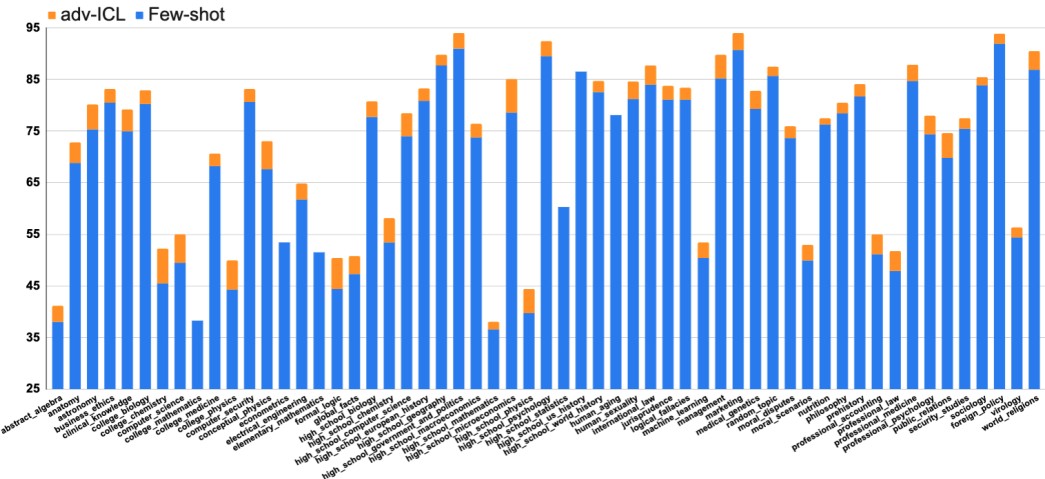

Figure 9: Results on MMLU using ChatGPT, where the y-axis begins at 25%, representing the baseline of random choices.

**Detailed results on BBH**    In Figure 10, we show the full results of ChatGPT on BIG-Bench Hard using 5-shot Chain-of-Thought prompting. The baseline achieves an average of 68.2% accuracy while adv-ICL reaches an average of accuracy of 70.6% and never performs worse than the baseline.

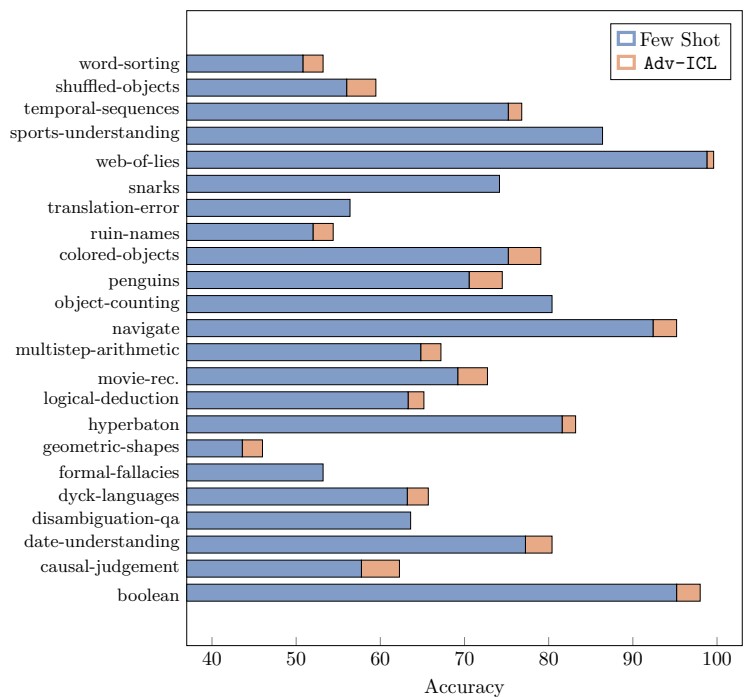

Figure 10: Full results on BBH using ChatGPT and 5-shot CoT prompting.

