# OpenReview forum: "Prompt Optimization via Adversarial In-Context Learning"
_ICLR.cc/2024/Conference — Submitted to ICLR 2024_

### Official Review · Reviewer_9rYe · 2023-10-31

**Soundness:** 2 fair
**Presentation:** 2 fair
**Contribution:** 2 fair
**Rating:** 5
**Confidence:** 4

**Summary:**

This paper proposes Adversarial In-Context Learning (adv-ICL) to optimize prompts for in-context learning (ICL).
It uses an LLM-based prompt modifier to modify the prompts of LLM-based generator G and LLM-based discriminator D.
In each round, the prompt modifier produces $r$ samples, and the best one is selected based on the discriminator loss to update the prompts of G and D.
The method shows improvements on 11 tasks.

**Strengths:**

1. The experimental results show clear improvements over the baselines.
2. Using adversarial optimization for prompt optimization is a novel idea.
3. The implementation and experimentation details are relatively adequate.

**Weaknesses:**

1. The novelty is limited:
    - The resampling method (as the prompt modifier) is already proposed in APE (Zhou et al., 2023). The prompt used to modify instructions in this work is very similar to the prompt used in APE but without proper citation.
    - For problems where the discriminator is trivial (e.g. for multiple-choice problems), the method is very similar to APE except that the demonstrations are also resampled.
    - The idea of adversarial optimization comes from GAN.
2. The presentation needs to be improved, I would suggest including a running example in the method section to explain the generator, discriminator, and prompt modifier, and how they interact with each other.
3. Some ablation studies are missing (see questions below).
4. The limitations are missing.

**Questions:**

1. Why the discriminator is necessary?
    - Can you do an ablation study when the discriminator is frozen?
    - Can you compare the performance of using a trivial discriminator (e.g. exact match) and using an LLM-based discriminator on classification and reasoning tasks?
    - Can you compare the performance of adv-ICL with APE which uses LLM's log-likelihood as the objective function, on summarization, data-to-text, and machine translation tasks?
2. In Table 3, too many iterations $T$ or samples $m$ seem to harm the performance, could you give some explanation or hypothesis about why?
3. In the appendix, for figures 9 and 10, what's the baseline in these figures? Is adv-ICL always not worse than the baselines and why?

Minor:
- Some captions could be longer to better explain the figures and tables.
- Please place the caption of the Tables above.

Misc:
- There is a concurrent work you may discuss in related work: LLM as optimizers [1].

---
The response partially addresses my concerns. I would raise my score from 3 to 5.

[1] Yang, C., Wang, X., Lu, Y., Liu, H., Le, Q. V., Zhou, D., & Chen, X. (2023). Large language models as optimizers. arXiv preprint arXiv:2309.03409.

---

> ### Author Response · Authors · 2023-11-20
> **Review response**
>
> We thank the reviewer for your time!
>
> > W1. The novelty is limited.
>
> To the best of our knowledge, this is the first work exploring optimizing discrete prompts for in-context learning (ICL) in the adversarial learning style. We propose adv-ICL which can effectively optimize the prompts. Despite simplicity, our method shows strong improvements over the baselines and achieves significant improvements over previous approaches while requiring very few training data.
>
> That being said, novelty itself is a subjective matter. The whole deep learning field is built on empirical results with old ideas. The whole GPT series had very limited “novelty” in the traditional sense, except the prompting formulation. They just made it work. Our method also works. Papers like ours are the reason for why we have great AI technologies today.
>
> > W2: Add examples in the presentation.
>
> We have updated an example in Figure 1 of the updated version. Please check it.
>
> > W4. The limitations are missing.
>
> The limitation section is not required for ICLR. Please check the author guide (https://iclr.cc/Conferences/2024/AuthorGuide).
>
> > Q1. Why the discriminator is necessary?
>
> We have conducted extra experiments supporting the reviewer’s comments.
>
> - Froze discriminator: We have conducted our experiments with Vicuna and ChatGPT where we frozen the discriminators on four representative datasets: WebNLG, Machine translation (RO->EN), YELP, and GSM8K. The results with Vicuna and ChatGPT are shown below:
>
> | Model | WebNLG | Translation (RO->EN) | YELP | GSM8K |
> | -------- | ------- | ------- | ------ | ------ |
> | Vicuna | 52.5 | 72.1 | 71.0 | 40.7 |
> | Vicuna with Adv-ICL (Frozen discriminator) | 54.9 | 71.9 | 71.2 | 41.8 |
> | Vicuna with Adv-ICL | 59.3 | 73.4 | 73.6 | 43.9 |
> |  |  |  |  |  |
> | ChatGPT | 60.9 | 78.8 | 69.8 | 79.4 |
> | ChatGPT with Adv-ICL (Frozen discriminator) | 60.8 | 78.6 | 70.1 | 79.5 |
> | ChatGPT with Adv-ICL | 63.6 | 80.4 | 71.9 | 82.3 |
>
> It is clear that updating the discriminator is essential in our formulation.
>
> > Q2. Why too many iterations T or samples m harm the performance
>
> This is an interesting point! We observed this phenomenon in the experiments and were also curious about it. Our current hypothesis is that, while in-context learning holds great promise, it does not behave similar to SGD based learning. For example, having more demonstrations or data samples in the prompt does not help. Also, as demonstrated by [1], a critical threshold exists for the number of training examples, and surpassing this threshold leads to a decline in performance. As a result, more iterations or more samples do not necessarily help.
>
> Given the complexity of in-context learning, we simply determine the best hyperparameters based on empirical results. We perform hyperparameter search based on the validation data of a small set of tasks. These hyperparameters do generalise to the other tasks. For details about the hyperparameter search process, you can refer to Section 2.4 in the original paper and Section 2.5 in the revised version.
>
> > Q3. Baseline in Figure 9 and Figure 10
>
> The “Baseline” in Figures 9 and 10 are the Few-shot baseline.
>
> > Misc: Discuss with concurrent work LLM Optimizers.
>
> We have included it in our related work discussion. The paper presents a novel LLMs optimizer that leverages feedback from LLMs to refine prompts effectively. This paper lies in the field of self-refinement and RL optimization m ethods, as explored in previous works [3] and [4]. However, our distinctive contribution lies in the study of adversarial in-context learning. We have included a discussion with the LLM optimizer in our related work.
>
> **References**
>
> [1] Min et al. Rethinking the Role of Demonstrations: What Makes In-Context Learning Work? ACl 2022.
>
> [2] Yang et al, Large language models as optimizers. arXiv preprint, Arxiv 2023.
>
> [3] Mingkai et al, Rlprompt: Optimizing discrete text prompts with reinforcement learning, EMNLP 2022.
>
> [4] Pan et al,  Dynamic prompt learning via policy gradient for semi-structured mathematical reasoning, ICLR 2023.

---

### Official Review · Reviewer_SGYz · 2023-10-31

**Soundness:** 1 poor
**Presentation:** 2 fair
**Contribution:** 2 fair
**Rating:** 3
**Confidence:** 4

**Summary:**

This paper proposes a new method called Adv-ICL that applies the framework of generative adversarial networks to LLMs to improve in-context learning. Instead of directly updating the weights of a target model, the method employ a prompt modifier to generate prompts, which are then fed into the generator and discriminator to lower their respective loss functions.

**Strengths:**

S1. The idea of applying generative adversarial networks to imporve in-context learning is plausible.
S2. According to the experimental results shown by the authors, Adv-ICL outperforms the baselines in some settings.

**Weaknesses:**

W1. The convergence of the proposed approach. It seem that the prompts are modified by chance rather than gradient signals. This poses a serious concerns on the convergence and efficiency of the proposed approach. Yet the paper does not discuss this aspect in details.

W2. Unconvencing Experiments:

W2-a. Disregarded Prompts:
During the iterative training process, when a new prompt is introduced to optimize the loss function, the model (either the generator or the discriminator) engages in in-context learning, regardless of whether the new prompt is accepted or discarded. It's worth noting that discarded prompts can also impact the model's training, but the paper does not explore this aspect in detail.

W2-b. Unclear Origin of Benefits:
The observed performance improvements in the experiments may not be solely attributed to the accepted examples but could also be influenced by the discarded ones. Furthermore, given that the optimization process relies on the stochastic behavior of large language models, it's possible that the learning effect results from the repeated trial-and-error of prompt modifications occurring during the training iterations rather than from improved prompts. Therefore, the assumption that improvements in experimental results are solely due to adversarial training may not be valid.

**Questions:**

1. Is there a method to modify the prompts that doesn't involve trial-and-error? How does it compared to gradient descent?
2. What are the training time and memory usage of your approach compared to the baselines?
3. Does your GANs framework converge effectively? What's the learning curves for the generator and discriminator, respectively?
4. How reliably can the experimental results be reproduced considering the stochastic nature of the optimization process and an LLM itself?
5. In terms of fairness, it seems that not all methods in the experiments use the prompts generated (or modified) by the same prompt modifier. Why?

---

> ### Author Response · Authors · 2023-11-20
> **Review response**
>
> We thank the reviewer for your time. We respectfully think that the reviewer did not understand our method. However, we will try to address your concerns as best as we can.
>
> > W1. The convergence of the proposed approach.
>
> In the updated draft, we included a new theoretical analysis section that shows the proposed algorithm converges to the desired equilibrium under mild assumptions.
>
> Here is the sketch of the analysis.
>
> ### **Theoretical Results**
> In this section, we theoretically analyze whether such a minimax objective in the form of in-context learning can achieve the desired equilibrium as in the original GAN scenario. We assume access to models with infinite capacities powering the discriminator $D$, generator $G$, and prompt modifier $M$ and that in each iteration, we sample a sufficient number of prompts from $M$ to update both $G$ and $D$. Let $p_{data}$ be the distribution of the training data, and $p_g$ be of the generated data from $G$.
>
> Considering a language model $\mathcal{M}$ which can be $D$ or $G$ performing a corresponding task $T_1$ and evaluated by a metric $E_1$,  we further assume that:
> 1. $M$ is powerful enough to modify the initial prompt of $\mathcal{M}$ for $T_1$, covering all possible prompt variants performing the task $T_1$.
> 2. $\mathcal{M}$ is a powerful enough language model that there exists a prompt $\mathcal{P}$ of $T$ for $\mathcal{M}$ that given $\mathcal{P}, \mathcal{M}$ can achieve the globally optimal result on $E_1$ for the task $T$.
> 3. There exists a prompt sampled by $M$ that maximizes $\mathcal{M}$ on $E_1$ globally on $T$.
>
> The assumption 3 is a result of the assumptions 1, and 2, and the assumption about our access to infinite capacities language models. Indeed, given $\mathcal{M}$, from assumption 2, there exists a globally optimized prompt $\mathcal{P}$ of $T_1$ for it such that it can achieve the globally optimal state on $E_1$ for the task $T_1$. Furthermore, since $M$ is powerful enough in modifying the initial prompt (ass. 1), plus $M$ samples a sufficiently large number of prompts for each iteration (ass. 2), $M$ can generate $\mathcal{P}$ with a non-zero probability, which conclude the assumption 3.
>
> From the assumptions, we have the following propositions.
>
> **Proposition 1 (Goodfellow et al., 2014).** For G fixed, the optimal discriminator D can be described in a closed form, denoted as D*.
>
> **Proposition 2.** For each training iteration, for a fixed $G$, the optimal discriminator $D^*$ can be achieved.
>
> **Proposition 3 (Motivated by Goodfellow et al., 2014).** If $G$ and $D$ have enough capacity, and at each training step, the discriminator is allowed to reach its optimum $D^*$ given $G$, and $p_g$ is updated so as to improve the criterion
>
> The proofs for the propositions are presented in *Appendix A.1* of our revised manual script. Our conclusion is that *with strong enough $D,G,M$, the framework adv-ICL converges*.
>
> > W2a. It's worth noting that discarded prompts can also impact the model's training
>
> We respectfully think that the reviewer misunderstood our method. First, it is worth noting that our method does not require any update in the models’ parameters, it only updates the prompts of the discriminator and generator. Second, we only forward the best-performing prompt from the previous training iteration to the next training iteration for modification, and it is reasonable for us to further optimize this best-performing prompt without considering the discarded ones.
>
> > W2b. Effect results may come from the repeated trial-and-error of prompt modifications
>
> In addition to our arguments in W2a, we further conduct experiments to verify whether the prompt modifier module worked as expected. Specifically, we deactivate the discriminator and only employ a prompt modifier to repeatedly optimize the prompt. Our experimental results with Vicuna and ChatGPT are provided below:
>
> | Model | WebNLG | Translation (RO->EN) | YELP | GSM8K |
> | -------- | ------- | ------- | ------ | ------ |
> | Vicuna | 52.5 | 72.1 | 71.0 | 40.7 |
> | Vicuna with Adv-ICL w/o Discriminator | 50.1 | 71.4 | 72.1 | 40.2 |
> | Vicuna with Adv-ICL | 59.3 | 73.4 | 73.6 | 43.9 |
> |  |  |  |  |  |
> | ChatGPT | 60.9 | 78.8 | 69.8 | 79.4 |
> | ChatGPT with Adv-ICL w/o Discriminator | 61.2 | 77.4 | 64.5 | 71.6 |
> | ChatGPT with Adv-ICL | 63.6 | 80.4 | 71.9 | 82.3 |
>
> In the majority of cases, when the discriminator is deactivated and only the prompt modifier is relied upon under Vicuna and ChatGPT, there is a noticeable decline in performance. This decline can be attributed to the inherent randomness of the prompt modifier, which lacks the ability to discern a more effective prompt. This observation underscores the substantial significance of the discriminator and adversarial loss in the optimization process.

---

> > ### Author Response · Authors · 2023-11-20
> > **Cont: Review response**
> >
> > > Q1. Avoid trial-and-error and adopt gradient descent.
> >
> > We have conducted experiments by incorporating the best-performing prompts from multiple previous iterations as feedback to refine the prompt in subsequent iterations (refer to our response to Reviewer iBKM Q1 for more information). However, when it comes to optimizing a discrete prompt, directly applying gradient descent is not feasible. In [5], the concept of 'gradient' in the context of a discrete prompt is defined, and our approach consistently outperforms it. For further details, please check Table 1 and Table 2 in the paper.
> >
> > In the case of a soft prompt such as [6] [7], gradient descent can be employed for optimization. Nevertheless, there is no universally applicable loss function. For instance, in tasks like summarization, reference-based metrics are ineffective for measuring performance accurately [8].
> >
> > Apart from all the pros and cons of gradient based approaches, the biggest problem is that it cannot be applied on state-of-the-art closed source models. We recognize the importance of doing experiments with open sourced models like Vicuna but the closed source models such as ChatGPT currently have better performance. So to know the effectiveness of our method, we need to perform experiments on closed source models.
> >
> >
> > > Q2. Training time and Memory usage
> >
> > Our method does not involve updating any parameters of the models and only relies on the inference state of LLMs. The training time for any models and settings is under 10 minutes, and the memory usage for vicuna-13b does not exceed 32GB.
> >
> > > Q3. Does your GANs framework converge effectively?
> >
> > Please refer to the response to W1 for more details.
> >
> > > Q4. Randomness
> >
> > We have rerun our experiments three times on WebNLG, RO->EN, YELP, and GSM8K. Below are our results with format run1/run2/run3:
> >
> > | Model | WebNLG | Translation (RO->EN) | YELP | GSM8K |
> > | -------- | ------- | ------- | ------ | ------ |
> > | Vicuna | 59.3/59.2/59.5 | 73.4/74.1/73.2 | 73.6/73.6/73.5 | 43.9/44.3/44.1 |
> > | ChatGPT | 63.6/63.5/63.8 | 80.4/80.6/80.6 | 71.9/71.8/71.9 | 82.3/82.5/82.2 |
> >
> > The results clearly demonstrate that Adv-ICL consistently delivers stable results, thereby highlighting its reliability in faithfully reproducing our experimental findings.
> >
> > > Q5. Not all methods in the experiments adopt the same prompt modifier.
> >
> > We conducted experiments with various baselines, each employing distinct prompt modification strategies. In order to make sure that we stay faithful to the originally proposed formulations, we used prompt modification strategies proposed by these techniques.
> >
> > **References**
> >
> > [1] Goodfellow et al. Generative adversarial nets. NeurIPS 2014.
> >
> > [2] Radford & Metz et al. Unsupervised Representation Learning with Deep Convolutional Generative Adversarial Networks. ICLR 2016.
> >
> > [3] Salimans et al. Improved Techniques for Training GANs. NeurIPS 2016.
> >
> > [4] Goodfellow. On distinguishability criteria for estimating generative models. ICLR 2015.
> >
> > [5] Reid et al. Automatic Prompt Optimization with "Gradient Descent" and Beam Search, Arxiv 2023.
> >
> > [6] Xiao et al. P-Tuning: Prompt Tuning Can Be Comparable to Fine-tuning Across Scales and Tasks, ACL 2022.
> >
> > [7] Xiang et al. Prefix-Tuning: Optimizing Continuous Prompts for Generation, ACL 2021
> >
> > [8] Zhang et al. Benchmarking Large Language Models for News Summarization. Arxiv 2023.

---

> ### Comment · Reviewer_SGYz · 2023-11-23
> **Thanks for your reply**
>
> The reviewer thanks the authors for their reply. I understand that the proposed approach does not alter the generator and discriminator. My major concerns were on the prompt selection and improvement parts in the prompt modifier, and how they can affect the overall convergence and training time of GANs. The reply did not address my concerns, so I will keep my original score.

---

> > ### Author Response · Authors · 2023-11-23
> >
> > Sure, thanks! Unfortunately we don't have time to further discuss details about the prompt selection part. Enjoy thanksgiving!

---

### Official Review · Reviewer_iBKM · 2023-11-01

**Soundness:** 3 good
**Presentation:** 3 good
**Contribution:** 3 good
**Rating:** 6
**Confidence:** 3

**Summary:**

In this paper, the authors introduce Adversarial In-Context Learning (adv-ICL), which uses 3 LLMs as a generator, a discriminator, and a prompt modifier to optimize prompts. Similar to traditional adversarial learning, there is a minimax game between the generator and the discriminator, where the generator aims to generate realistic text to fool the discriminator. The generator is provided with task instructions, several exemplars, and input at each round and produces the output; the discriminator then tries to decide whether the generator input-output pair is real data or model generated. After that, the prompt modifier makes changes to the generator, and the discriminator prompts to improve the adversarial loss. They perform a set of experiments using both open and closed-source models on various generation and classification tasks to evaluate their model.

**Strengths:**

- Since there are no updates to the model parameters and only the prompts change, adv-ICL is computationally efficient and effective in low-resource settings. Moreover, adv-ICL only needs a few iterations and training samples in order to achieve high performance.

- There is a thorough analysis of the quantitative and qualitative aspects of their method.

**Weaknesses:**

- Some more RL-based prompt optimization baselines (e.g., Mingkai Deng, Jianyu Wang, Cheng-Ping Hsieh, Yihan Wang, Han Guo, Tianmin Shu, Meng Song, Eric P. Xing, & Zhiting Hu. (2022). RLPrompt: Optimizing Discrete Text Prompts with Reinforcement Learning.) could be used in the evaluation section to provide more insight.

**Questions:**

- It seems that in order to make edits to the prompts, the prompt modifier is prompted with a template text, and the last best-performing generator and discriminator prompts. What if you provide more feedback to the prompt modifier, such as the last best-performing prompt, alongside its predecessor, and how much better this last prompt performed than the other?

- For human evaluation, it is mentioned that annotators are tasked with verifying whether the sampled instruction/demonstration is semantically similar to the original one or not. What if you also add a module that can automatically verify this, check the content preservation score, and also consider this metric when choosing a prompt?

---

> ### Author Response · Authors · 2023-11-20
> **Review response**
>
> We thank the reviewer for your time.
>
> > W1. Need to be compared with some more RL-based prompt optimization baselines.
>
> These RL-based baselines could not be applied to the closed source models such as ChatGPT and text-davinci. To test the effectiveness of our method, we need to perform experiments on these models because they have better performance. This is why we made the decision to not compare with RL-based methods. Specifically, [1] and [2] require access to LLMs for inserting MLPs.
>
> We compared adv-ICL with state-of-the-art prompt learning methods including GPS, APO and APE and outperformed them consistently, which indicates the effectiveness of our method.
>
> > Q1. What if we provide more feedback to the prompt modifier.
>
> We conducted an experiment that involved integrating the most successful prompts from previous iterations as feedback for the next iteration. In this process, we utilized previous best-performing prompts, namely $P_1, P_2, ..., P_k$, as inputs to the prompt modifier module when generating the $(k+1)$-th prompt.
>
> We applied the method to four representative tasks, including data-to-text generation task WebNLG, machine translation, sentiment classification Yelp, and reasoning GSM8k, using Vicuna and ChatGPT. The experimental results are shown below:
>
> | Model | WebNLG | Translation (RO->EN) | YELP | GSM8K |
> | -------- | ------- | ------- | ------ | ------ |
> | Vicuna | 52.5 | 72.1 | 71.0 | 40.7 |
> | Vicuna with Adv-ICL (prompt modifier with history) | 56.9 | 74.0 | 74.2 | 42.2 |
> | Vicuna with Adv-ICL | 59.3 | 73.4 | 73.6 | 43.9 |
> |  |  |  |  |  |
> | ChatGPT | 60.9 | 78.8 | 69.8 | 79.4 |
> | ChatGPT with Adv-ICL (prompt modifier with history) | 62.1 | 79.8 | 72.1 | 80.9 |
> | ChatGPT with Adv-ICL | 63.6 | 80.4 | 71.9 | 82.3 |
>
> In the case of Vicuna, incorporating additional feedback into the prompt modifier proves effective for tasks such as translation and classification. However, this approach falls short when applied to data-to-text and reasoning tasks. On the other hand, for ChatGPT, augmenting the prompt modifier with more feedback does not yield improved performance.
>
> > Q2. What if you also add a module that can automatically verify modifications.
>
> As shown in the “Human evaluation of prompt modifier performance” part of Section 3.3, most of the prompts outputted by the prompt modifier are semantically correct. Specifically, the correct percentage of  text-davinci-002, ChatGPT, and Vicuna are 88%, 91%, and 83% respectively. So we do not think that it is necessary to automatically verify modifications.
>
> **References**
>
> [1] Mingkai et al, Rlprompt: Optimizing discrete text prompts with reinforcement learning, EMNLP 2022.
>
> [2] Pan et al,  Dynamic prompt learning via policy gradient for semi-structured mathematical reasoning, ICLR 2023.

---

> > ### Comment · Reviewer_iBKM · 2023-11-23
> >
> > Thank you for the explanations and for providing the results of your additional experiments. I have changed my overall score from 5 to 6.

---

> > > ### Author Response · Authors · 2023-11-23
> > >
> > > We are glad that you find our explanations helpful. Appreciate the feedback. Thanks!

---

### Author Response · Authors · 2023-11-20
**General Reply**

The results of this rebuttal and academic research in general seem so insignificant given what’s happening in the industry, amidst the time of great changes. OpenAI rose so quickly. And it could fall. But this is exactly the time when research is never more important. It was due to people like Alec Radford and Geoffrey Hinton who worked on research single-heartedly that we had today’s success. So we should do the same. Look for things to believe in, do research to test our belief, and repeat.

Coming back to the reviews, we thank the reviewers for their time. But we feel that our paper did not receive an adequate judgement. We received 3 reviews and 2 of them do not reflect a fair judgement of our work. Particularly, reviewer iBKM did not point out any intrinsic weakness of our work but gave a score of 5. Reviewer SGYz did not understand our method and gave a score of 3. All academic people joke about the randomness of the review process yet we are both the offenders and victims.

That being said, we do find some of the reviews insightful and helpful. So we address the reviewers’ questions and concerns as follows:

1. We added a theoretical analysis about the convergence properties of adv-ICL. Basically, our analysis shows that the algorithm does converge to the original conditioned GAN scenario given mild assumptions. Please check Section 2.4 in the updated draft for more details.
2. As requested by reviewer iBKM, we updated results using more feedback to the prompt modifier.
3. As requested by reviewer SGYz and reviewer 9rYe, we updated results with the discriminator removed or frozen.
4. As requested by reviewer 9rYe, we studied the variances of the method.
5. We made a few improvements to the presentation of the paper.

Please refer to the highlighted parts for the updated content. Some of them are in the appendix.

Overall, we feel that we need a fairer evaluation. And we do our best to address reviewers’ questions and concerns. Please do not hesitate to let us know if you have any further questions.

---

### Meta-Review · Area_Chair_DsZ5 · 2023-12-08

**Metareview:**

The paper proposes Adversarial In-Context Learning (adv-ICL), a method that employs adversarial learning concepts to optimize prompts for in-context learning using LLMs. The paper's strengths and weaknesses, as noted by the reviewers, are summarized below:

Strengths:

+ Computational Efficiency: Adv-ICL's reliance solely on prompt modifications, without updating model parameters, makes it computationally efficient and particularly effective in low-resource settings.

+ Novel Application of GANs: The idea of applying generative adversarial networks (GANs) to improve in-context learning is innovative and shows potential based on the experimental results.

+ Performance Improvements: According to the experiments, adv-ICL outperforms baselines in some settings, demonstrating clear improvements.

+ Comprehensive Analysis: The paper includes a thorough analysis of the quantitative and qualitative aspects of the method.

Weaknesses:

- Limited Novelty: The novelty of the approach is questioned, particularly in relation to the resampling method used as the prompt modifier, which appears similar to previously proposed methods in existing literature.

- Presentation and Clarity Issues: The presentation of the method and its components lacks clarity. Inclusion of a running example to explain the interactions between the generator, discriminator, and prompt modifier would improve understanding.


- Convergence Concerns: There are serious concerns about the convergence and efficiency of the proposed approach, as the prompts seem to be modified by chance rather than clear gradient signals. (Addressed in the rebuttal)

- Unconvincing Experiments and Lack of Detail: The paper fails to convincingly demonstrate that performance improvements are solely due to the adversarial training and not influenced by stochastic behaviors or discarded prompts. (Partially addressed in the rebuttal)

Given these observations, the paper, while presenting an interesting and potentially impactful idea, falls short in several areas.

**Justification For Why Not Higher Score:**

See weaknesses

**Justification For Why Not Lower Score:**

N/A

---

### Decision · Program_Chairs · 2024-01-16

Reject